# Modulating Elements of Nurse Resilience in Population Care during the COVID-19 Pandemic

**DOI:** 10.3390/ijerph19084452

**Published:** 2022-04-07

**Authors:** Ester Sierra-García, Eva María Sosa-Palanca, Carlos Saus-Ortega, Antonio Ruiz-Hontangas, Raúl Juárez-Vela, Vicente Gea-Caballero

**Affiliations:** 1Nursing School La Fe, Adscript Center of Universidad de Valencia, 46026 Valencia, Spain; estersigar@gmail.com (E.S.-G.); saus_car@gva.es (C.S.-O.); 2Research Group GREIACC, Health Research Institute La Fe, Hospital La Fe, 46016 Valencia, Spain; 3Faculty of Health Science, European University Valencia, 46010 Valencia, Spain; antonio.ruiz2@universidadeuropea.es; 4Department of Nursing, University of La Rioja, 26006 Logroño, La Rioja, Spain; raul.juarez@unirioja.es; 5Faculty of Health Science, International University of Valencia, 46002 Valencia, Spain; vagea@universidadviu.com; 6Research Group Patient Blood Management PBM, Health Research Institute IdIPAZ, Hospital La Paz, 28046 Madrid, Spain

**Keywords:** nurse, COVID-19, resilience, coping, stress, uncertainty

## Abstract

COVID-19 has significantly affected the work environment of nurses. In the face of the challenges posed by stressors in clinical practice, some nurses adapt and prove to be resilient. In the face of the COVID-19 pandemic, the nature of care itself and the new ways of working are potentially very stressful. We aim to analyze the resilience of care nurses to the psychological impact of the COVID-19 pandemic. This study is a systematic review of nurse caregiver resilience to the COVID-19 pandemic in 2021. Our search was conducted in the WOS, Medline/PubMed, Cochrane, BVS/LILACS, and Cuiden databases. The inclusion criteria were: studies published in Spanish or English; carried out from March 2020 to May 2021 on nurses caring for patients with COVID-19; and investigating the factors influencing the psychological impact, resilience, strategies to develop it, and interventions to promote it during this pandemic and others, such as SARS, MERS, or ebola. The quality of the studies and the risk of bias were evaluated following ICROMS, STROBE and AMSTAR-2 criteria. Twenty-two studies were selected. Most of the studies highlighted the presence of stressors in nurses, emphasizing those of the environment, which converged in dysfunctional responses that hurt their resilience. The most persuasive factors were social and organizational support. Coping strategies developed by nurses and especially interventions by organizations were detected as instruments to foster resilience, but have not been well researched. Resilience has a key moderating role in mitigating the psychological impact of nurses in the face of the COVID-19 pandemic.

## 1. Introduction

In 2018, the International Council of Nurses (ICN) and the World Health Organization (WHO) launched the Nursing Now movement to raise the status and profile of nurses, capping it off with the year 2020 [1]. The iconic year of nurses sees the emergence of COVID-19, an emerging infectious disease caused by the new SARS-CoV-2 coronavirus, which was first identified in Wuhan on 31 December 2019 [2]. The WHO declared the disease a pandemic on 11 March 2020 due to its rapid spread in most countries of the world [3]. COVID-19 has been a dramatic global disruption, as the global number of deceased and infected patients with this disease was 4.7 million and 229.6 million, respectively, as of September 2021 [4]. The latest ICN analysis shows that as of November 2020, the number of nurses who have died after contracting COVID-19 is 1500, as many as in World War I [5]. This figure, which includes nurses from only 44 of the world’s 195 countries, is known to be a low estimate of the actual number of deaths.

Nurses are particularly at risk of psychological problems due to the highly stressful work situations to which they are exposed; this fact can be more challenging when applied in the context of a pandemic, as working in the face of such a situation can be exhausting [6]. Currently, nurses have not only experienced an increase in the volume and intensity of their work [7] but have to adapt to new protocols and new normality. A Spanish study showed that one in seven healthcare workers tested positive for a disabling mental disorder during the first wave of the COVID-19 pandemic, such as major depressive disorder, generalized anxiety disorder, panic attacks, and post-traumatic stress disorder [8]. In previous epidemics or pandemics [9,10], nurses presented the highest levels of occupational stress and distress compared to other personnel involved, indicating that they may have subsequently developed mental disorders. However, in the face of the negative consequences of stressors in clinical practice, there are nurses who demonstrate resilience after being exposed to a traumatic event and are successful in the face of the same challenges and limitations [11]. The American Psychological Association [12] defines resilience as the ability to adapt to adversity, trauma, tragedy, threats, or other significant sources of stress and, from this, seek a drive to cope and emerge successfully.

From the nursing discipline and perspective, this concept is related to Dorothy Johnson’s Behavioral Systems Model, which focuses on how present or potential stress can affect a person’s ability to adapt. For Johnson, the person is an interactive, interdependent, and integrated behavioral system with patterned, repetitive, and intentional forms of behavior [13,14]. The author reflects in her model that the human system is constantly subjected to stressors [14,15,16], which correspond to internal or external stimuli that produce tension and a certain degree of instability, giving rise to constructive or destructive behavioral changes, which can lead to equilibrium or imbalance. In view of this, the objective of the model is to maintain and recover equilibrium, helping the person to achieve an optimal level of functioning. This is why it is directly related to the term resilience [13], which is being increasingly researched within the field of nursing, since it is a capacity that is influenced by various factors and which allows the person to continue to project him/herself despite destabilizing events, both in patients and in the nursing staff itself. It has been shown that the most resilient nurses can better tolerate the hostile environment of pain, suffering, and death that they encounter throughout the workday [17], an environment that is especially aggressive during the pandemic.

Given the continuous exposure of nurses to human suffering and an adverse work environment, resilience has become so important that it is considered an inherent characteristic of nurses during the course of healthcare; there is evidence that resilience ameliorates the effects of work stress and largely avoids its long-term consequences [13]. Resilient nurses are seen as a crucial element in an ever-changing healthcare system [18], especially during the new SARS-CoV-2 pandemic situation.

In view of the above, it is relevant to explore care nurses’ resilience in the face of the impact of the COVID-19 pandemic. Researching this could allow us to identify which aspects modulate its development, obtaining information on those nurses who prosper and continue to find satisfaction in their profession despite the current challenges and problems posed by this pandemic. This study aims to answer the research question: “Does resilience modulate the psychological impact on nurses of the COVID-19 pandemic?”. Additionally, therefore, the aim of this study is to analyze the resilience of care nurses to the psychological impact of the COVID-19 pandemic.

## 2. Materials and Methods

### 2.1. Design

Systematic review was conducted between March 2020 and May 2021 according to PRISMA guidelines [19].

### 2.2. Protocol Development and PICO Question

The review protocol was developed to meet PRISMA [19], which was designed to answer PICO (Population, Intervention, Comparison, Outcome).

However, the stated focus question, “Does resilience modulate the psychological impact of nurses in the face of the COVID-19 pandemic?” was an adaptation of a PIO format question (Population, Intervention, Outcome) [20]: nurses (P), resilience (I), and psychological impact in the face of the COVID-19 pandemic (O).

The protocol for this review was not registered due to the rush to collect data in the midst of the pandemic and the desire to share our findings quickly.

### 2.3. Selection Criteria

#### 2.3.1. Inclusion Criteria

We included studies published in Spanish or English; conducted (from March 2020 to May 2021) on nurses who cared for patients with COVID-19; investigating the influential factors of the psychological impact, resilience, strategies to develop it, or interventions to promote it during this pandemic and during others, such as SARS, MERS, or Ebola.

#### 2.3.2. Exclusion Criteria

We excluded studies conducted on pediatric nurses, literature reviews, and papers that did not meet the necessary methodological quality.

### 2.4. Search Strategy

An exhaustive search was initiated in different databases. Electronic databases: Web of Science, Medline/Pubmed, Cochrane Collaboration (The Cochrane Library), BVS Biblioteca Virtual en Salud/LILACS, and CUIDEN. Subsequently, a manual review of the bibliographic references of the selected articles was carried out in order to include other potentially valid studies for the review. The keywords used concerned Medical Subject Headings (Mesh) [21]: nursing/nurses; coronavirus infections; resilience, psychological; adaptation, psychological; burnout, psychological; stress, psychological; uncertainty. In order to combine these terms, AND and OR were used as Boolean operators. Truncation operators were used as accuracy operators. Two researchers reviewed the selected and eliminated papers; if there were any discrepancies, a third reviewer intervened to decide their inclusion or exclusion.

The search strategy consisted of elaborating different search strings based on the above descriptors and free text (Table 1).

### 2.5. Study Variables

The variables in the review were: influential factors on the psychological impact of the COVID-19 pandemic, behavioral responses exhibited by nurses, level of nurses’ resilience, influential factors on observed resilience, resilience-development strategies, and healthcare organization interventions to promote resilience in professionals.

Documentary quality assessment:

Documentary quality was assessed through levels of evidence of effectiveness according to the Joana Briggs Institute (JBI) [22]; these are designed to align with the GRADE approach to pre-ranking findings based on the study design, which are then upgraded or downgraded depending on a number of factors.

Regarding the tool used to assess the methodological quality and the risk of bias of the articles, ICROMS [23] was used; it brings together different quality and methodological bias tools in a single document, allowing the quick and efficient selection and evaluation of different aspects and indicators of methodological quality. The tool consists of two parts: a list of quality criteria specific for each study design, as well as criteria applicable across all study designs by using a scoring system, and a "decision matrix", which specifies the robustness of the study by identifying minimum requirements according to the study type and the relevance of the study to the review question. The decision matrix directly determines inclusion or exclusion of a study in the review. ICROMS was used to analyze the quality and bias of the qualitative studies, whose minimum score was 16.

To assess the quality of systematic reviews, the AMSTAR-2 [24] tool was used, which was developed to evaluate systematic reviews. It allows a more detailed evaluation of the SRs that also include non-randomized studies of health interventions, which are increasingly incorporated in the SR. This questionnaire contains 16 domains with simple response options: “yes”, when the result is positive; “no”, when the standard was not met or there is insufficient information to answer; and “partial yes”, in cases where there was partial adherence to the standard. Amstar-2 identifies the quality of reviews as high, moderate, low, or critically low quality.

Quality assessment of the cross-sectional studies in this review was performed using the STROBE [25] tool. The STROBE Statement is a list composed of 22 points that are considered essential for adequate communication of observational studies, allowing a critical evaluation of them. The cutoff score was 18/22.

## 3. Results

The total number of records obtained at the end of the literature search was (*n* = 1578), of which 22 were finally selected. The selection process can be seen in Figure 1, which was performed by two independent reviewers. The bibliographic search and the selection of documents are described in Figure 1. A total of 22 studies were chosen to be included in this review. Regarding the design of the studies that formed part of this systematic review, seventeen cross-sectional descriptive observational studies [26,27,28,29,30,31,32,33,34,35,36,37,38,39,40,41,42,43], one mixed-methods systematic review [44], one meta-analysis and systematic review [45], one scoping review [46], and one qualitative study [47] were identified.

The factors that predominated with respect to the influence of the psychological impact of nurses were those related to gender [27,31,32,39,45], age [28,31,32,38,41,43], work experience [29,31,33,35,36,43], family [27,31,38,41], working conditions, use of personal protective equipment, degree of training [27,35,36,37,38], socioeconomic level, social support and organizational support [35,36,37,39].

An increase in tension and stress among nurses was detected compared to before the pandemic [33]. The most notable responses included fear, helplessness, discomfort with the scarcity and prolonged use of PPE, emotional distress, perception of providing poor nursing care [29,31,36,38,42,45], and concern and uncertainty for their safety and that of their families [29,36,38,40,42], even to the point of considering leaving the profession [36,38]. Physical symptoms related to the psychological burden were also detected [31,32,36,41].

About the level of resilience, the research used different validated scales to measure this variable, such as the Connor–Davidson Resilience Scale (CD-RISC) [26,27,29,33,34,36,37,40,42] and the Brief Resilient Coping Scale (BRCS) [35,38]. A total of eight articles referred to low–moderate levels of resilience in nurses [26,29,33,34,35,36,37,38], and three studies reported high levels [27,40,42], also alluding to the impact of resilience on the quality of care provided to patients with COVID-19 [32,34,35,45].

All studies highlighted the protective role of resilience in the psychological impact of the pandemic on these professionals. The factors that influenced the resilience of the nurses facing this outbreak were: organizational support [26,27,31,32,34]; the perception of greater social support [27,32,35,45]; having training, work, and emergency experience [26,30,31,35,45]; having an optimistic, tenacious, and confident attitude [33,45]; and sociodemographic variables, such as being older, being male, and having a medium–high socioeconomic level [30,31,43].

Among the strategies developed by the nurses to favor their resilience and mitigate the psychological impact, the following prevailed: seeking socio-familial support [28,38,42,43], carrying out recreational and health-promoting activities [28,42,44], seeking information sources on psychological resources and clinical practice [29,42,43], and developing a positive and tenacious attitude [38,42,43,44,45]. These strategies helped to stabilize the nurses emotionally and strengthen their resilience. The strategy they used the least was seeking professional psychological support [29,30,38,42].

Research indicated that nurses tended to adopt positive strategies in the face of the psychological impact of the COVID-19 [43] pandemic [28,29,38,42]. It was found that women were less likely to use coping strategies [28,29] and that nurses who suffered stress due to insufficient preparation and fear of contagion did not establish adequate strategies [29,33].

Concerning interventions to favor resilience and reduce the psychological impact of nurses in the face of this pandemic, the following were highlighted: offering training and clear instructions on the approach to patients with COVID-19, improving the working environment and working conditions, providing optimal protective material, promoting basic needs, and psychoemotional management [31,34,38,39,42,46,47]. Two articles classified interventions in terms of temporality, i.e., those to be carried out before exposure to the outbreak [42,46] (such as the provision of optimal education and training for the management of patients’ psychological problems, diagnosis, and treatment; guidelines for infection prevention and control; resilience training to create a sense of preparedness for clinical practice); and during exposure [42,46] (such as providing personal protective equipment; favoring a good working environment; improving resilience through the adequate provision of information, psychosocial support, and treatment; monitoring the health status of professionals; and using various forms and contents of psychosocial support). Several barriers and guidelines were identified for the implementation of interventions to support nurses’ resilience [44], such as the lack of awareness of practitioners’ needs by organizations and limited resources, including a lack of materials, time, and staff skills.

The synthesis of the articles with the most relevant information from the retrieved documents can be seen in Table 2.

## 4. Discussion

This review was based on Dorothy Johnson’s Behavioral Model [13,14,15,16,48] to broadly and holistically analyze the resilience of nurses in the face of the impact of the COVID-19 pandemic, which was a key competence to reducing or controlling the effects of the stress generated by this catastrophe. To this end, we explored the factors that influenced the psychological impact of the nurses, categorized by Johnson as internal stressing factors (ISF) and external stressing factors (ESF) [14], the behavioral and psychological responses in which they converged, as well as the determination of the nurses’ resilience levels and the factors that influence it, classified as intrapersonal resilience factors (IRF) and environmental resilience factors (ERF), among other factors studied (Figure 2).

As in the current study, a high prevalence of stressors led to responses leading to psychological imbalance and distress in nurses in previous outbreaks, such as Ebola, H1N1, and SARS [49,50,51,52]. Evidence showed that female nurses had a significant risk of psychological distress, consistent with previous findings [49,50,51,52,53]. In this sense, nurses generally tend to be predominantly female, thus presenting higher additional workloads [54]. Being a nurse idiosyncratically implies having close contact with patients with COVID-19 and, therefore, a higher risk of becoming infected [33,38,39,43]; this aspect was identified as a key ESF for nurses in the context of this outbreak, as it has implications not only for their own health but also for that of their families [55].

It was detected that the preponderance of ESF has primarily induced dysfunctional psychological responses [33,36,38,39,42,43] in the affiliation, dependence, fulfillment, and aggression/protection subsystems, as well as dysfunctional physical responses [28,35,41,43] in the ingestion, elimination, and sexual subsystems, generating an imbalance in the nurses. This relates to Johnson’s theory that a person’s attempts to regain equilibrium in the face of a powerful force may require extraordinary energy consumption, even affecting biological processes [14]. These findings are in line with experiences in previous epidemics [52,56,57] and may indicate that mental disorders may develop in the long term.

The fundamentally low–moderate levels of resilience detected in nurses in the review [26,29,33,34,35,36,37,38] were consistent with research conducted in other outbreaks, where concern was expressed about their low resilience [58,59,60,61]. Even resilient nurses were found to be stressed, albeit to a lesser degree. This fact is consistent with Jonhson’s postulation that balance can also occur in illness [16]. Therefore, being resilient does not mean that nurses cannot experience feelings of emotional distress but rather that they have the skills to adapt to such a situation.

Nevertheless, the three studies that reported high levels of resilience [27,40,42] agreed that the aspects that marked this difference for other nurses were primarily related to ARF. This was in line with Johnson’s approach, influenced by F. Nightingale, on the importance of approaching the person by focusing on his or her relationship with the environment, not with the disease [48]. The evidence was unanimous in indicating that the perception of social support was linked to less distress [29,32,33,34,35,41], as this variable is an ERF known to be effective in reducing stress among nurses [13,62,63]. They especially valued social support from their families, a fact that is compatible with data from research on resilience. This may be the case because an individual’s first social relationship is usually with his or her family. In the COVID-19 pandemic, this type of support was scarce [29,33,35,41], probably due to the isolating conditions caused by this virus, in addition to the fact that many were rejected by family and friends. Nurses not only had to deal with the COVID-19 pandemic but also with their concern for their families and the stigmatization of the public [31,41], so they had limited access to one of the key elements to foster their resilience, which in this study was a relevant ESF.

Because of this, organizational support from governments, hospitals, supervisors, and coworkers is especially important to fill this gap, being another ERF with an important positive influence on resilience [26,30,32,39,43]. Some studies state that higher levels of organizational support are significantly associated with effective work outcomes, fostering organizational identity, positive work attitudes, satisfaction, job commitment, and improved physical and mental health of nurses [62,64]. However, several articles in this review [26,29,33,34,35,36,38] in which nurses had lower levels of resilience coincided with reporting a deficit of organizational support, making it a relevant ESF. Therefore, it was detected in this research that the fluctuation of resilience levels was significantly influenced by the presence of ERF.

The coping strategies developed by the nurses denoted a mediating role between resilience and stress. They were an effective tool to develop and reinforce IRF [27,28,30,31,37,41,43] and, at the same time, act primarily against EIFs since, due to the peculiarities of the transmission of this virus, the nurses had limited access to ARFs. Previous research has shown that problem-focused strategies are positively related to indicators of resilience, decreased stress, and psychological distress [65,66]; however, evidence from this study showed the opposite. This could occur because aspects such as the effort made to control the problems related to this virus, being permanently informed about it, and the fear of infection presented a relationship with greater stress, apart from the fact that they, in turn, were ISF detected in this review. Studies before the COVID-19 pandemic also concluded that emotion-focused strategies presented more inconsistent results in their relationship with nurses’ well-being [66,67,68], but in this research, they presented a stronger connection with resilience. This could be because these types of strategies included, for example, adopting a positive attitude, performing relaxation techniques, or performing health-promoting activities [27,30,31,37,41,43], which were linked to a pragmatic way of embodying nurses’ IRF.

Through this review, a group was revealed that requires special attention due to its greater vulnerability: nurses who are female, younger, and with less work experience [28,29,33] (three variables corresponding to ISF detected in this study). This could be related to the fact that many of this group are assigned jobs in special services or in areas of higher acuity/gravity. Additionally, some countries accelerated their final-year nursing students to join the nursing team earlier [69]. Mental health screenings would entail an approach that could help monitor both nurses’ distress and the use of appropriate coping strategies. In fact, in this review, several studies [29,38,42] agreed that the strategy least used by nurses was seeking professional psychological support. We believe that this is due to the fear of being stigmatized for the use of this type of resource and reluctance to acknowledge the need for help.

Although relatively few studies investigated organizational interventions to promote nurses’ resilience in the face of this pandemic, it is worth noting that the findings were extremely consistent, as the studies where higher levels of resilience were reported were those where organizational interventions were applied [27,40,42]. This lends confidence to the suggestion that implementing IRF and ERF through interventions are important targets for reducing the imbalance generated by both ISF and ESF and could even contain them and strengthen resilience. It may be too early to achieve published work in this area, but in other outbreaks or epidemics, common or similar barriers and facilitators were developed [44]; some research in other epidemics [70,71,72,73] suggests that providing education, training in resilience, providing psychological support, and providing the necessary resources are aspects that increase resilience and a sense of empowerment among nurses. This is consistent with previous experiences, in which it is reported that nurses repeatedly feel ignored by their managers when they raise concerns about their mental health [74].

Evidence from this study identified several barriers to the implementation of interventions by organizations, such as their lack of awareness of health workers’ needs, along with resource constraints, including a lack of equipment, time, and staff skills. Organizational resilience [75] is defined as the ability of an organization to anticipate, prepare for, respond to, and adapt to incremental changes and sudden disruptions to survive and thrive, so this study shows that the impact of COVID-19 has been an additional assault on nurses’ resilience, as nurses already had stressors in the healthcare environment before the pandemic [44,76]. The pandemic has also influenced the resilience of organizations, which was already impaired [75]. If organizations are weak, they will not be resilient and therefore will not be able to effectively develop interventions to develop competence in nurses, who in turn are part of these structures. Increasingly, evidence supports the close relationship between organizational resilience and outcomes [77], as nursing resilience must be approached as a collective and organizational responsibility, as it is not built alone, nor is it solely an individual responsibility.

Among the limitations found in this review, in addition to those of each of the papers reviewed (which, due to their design, do not generate a high level of evidence), it should be noted that in several cases, the interventions carried out by the organizations were communicated through brief comments, letters to the editor, editorials, or communications at congresses, probably driven by the urgent desire to share the findings during the period of the health crisis. By discarding these sources of information from the selection criteria, confidence that all relevant research was found is reduced.

The study results were collected in the most critical period of the pandemic, and therefore, some studies published later may not have been included in this review. Some psychosocially focused databases were not used, as a more clinical approach was intended; this may also have caused some articles to be missed. On the other hand, given that during the worst moments of the COVID-19 pandemic, scientific production was necessarily rapid and agile, it is logical that many cross-sectional observational articles were published, which we decided to include in the study despite the low evidence they provide. To evaluate them, we chose the STROBE tool, which, although it does not evaluate bias, is one of the tools used in Cochrane and is commonly used in published reviews, also of a systematic type.

## 5. Conclusions

Resilience has a key moderating role in mitigating the psychological impact of the COVID-19 pandemic on nurses in care, promoting their equilibrium, and promoting the quality of their work.

The COVID-19 pandemic has provoked massive exposure to stressors in nurses; external stressors are the main contributors to the responses generated, converging in those of psychological and physical etiology.

The levels of resilience of nurses in the face of this health crisis are predominantly low–moderate, whose fluctuation is due to the interaction between intrapersonal and especially environmental factors, along with the presence of stressors.

The strategies developed by nurses to cope with the psychological impact are useful to promote IRFs, partially promote ARFs, and combat SIFs. Identifying when nurses may possess factors that expose them to developing negative coping strategies, ineffective coping, and finding ways to foster more positive approaches to managing stress will be key to nurses ultimately achieving optimal levels of resilience while avoiding imbalances and the risks of adverse health consequences.

Interventions developed by healthcare organizations play the most relevant role in fostering nurses’ resilience in the face of this pandemic, as it is through them that this skill can be holistically and comprehensively trained and promoted.

## Figures and Tables

**Figure 1 ijerph-19-04452-f001:**
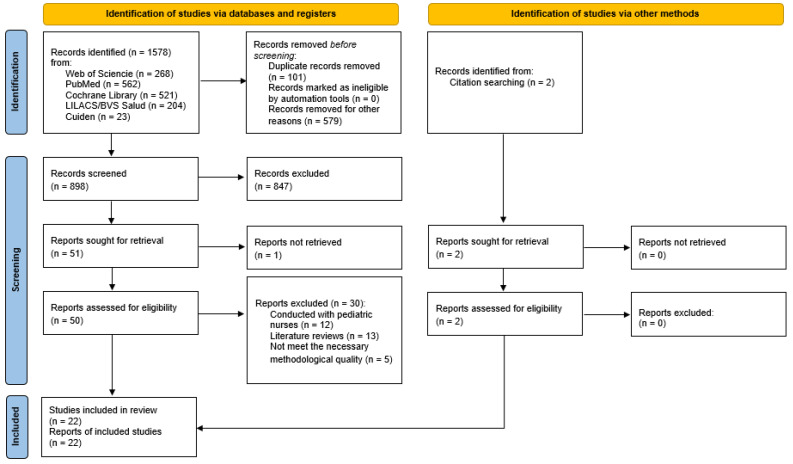
Flow chart.

**Figure 2 ijerph-19-04452-f002:**
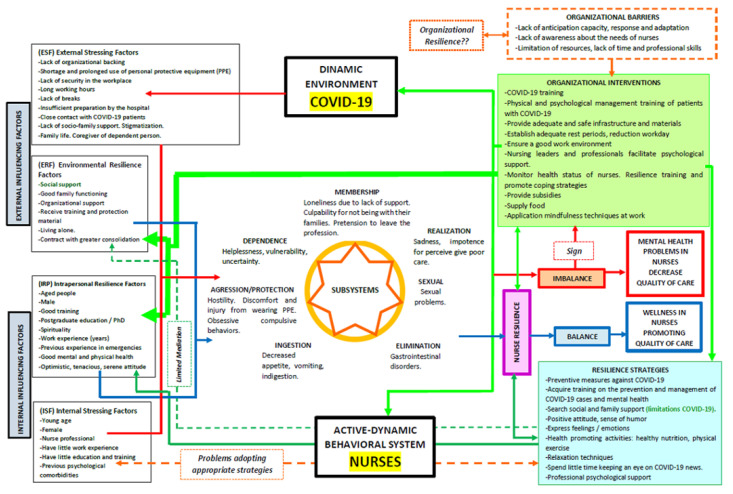
Intrapersonal resilience factors, environmental resilience factors, and other factors studied.

**Table 1 ijerph-19-04452-t001:** Search strategy.

Database	Search
WOS	((nurs *) AND (COVID-19)) OR (coronavirus infections)) AND (Resilience, Psychological)) OR (Adaptation, Psychological)) OR (Stress, Psychological)) OR (Burnout, Psychological)) OR (uncertainty)
Medine/ PubMed	((nurs *) AND (coronavirus infections)) AND (resilience, psychological) Filters: in the last 1 year
	((nurs *) AND (coronavirus infections)) AND (adaptation, psychological) Filters: in the last 1 year
	((nurs *) AND (coronavirus infections)) AND (stress, psychological) Filters: in the last 1 year
Cochrane Library	((nurs *) AND (coronavirus infections)) AND (resilience, psychological)
	((nurs *) AND (coronavirus infections)) AND (adaptation, psychological)
	((nurs *) AND (coronavirus infections)) AND (stress, psychological)
LILACS/ BVS Salud	(nurs *) AND (coronavirus infections) AND (resilience, psychological) AND (year_cluster: [2020 TO 2021])
	(nurs *) AND (coronavirus infections) AND (adaptation, psychological) AND (year_cluster: [2020 TO 2021])
	(nurs *) AND (coronavirus infections) AND (stress, psychological) AND (year_cluster: [2020 TO 2021])
Cuiden	(“enfermera”) AND ((“COVID-19”) AND (“resiliencia”))
	(“enfermera”) AND ((“COVID19”) AND (“afrontamiento”))
	(“enfermera”) AND ((“COVID-19”) AND (“adaptación”))
	(“enfermera”) AND ((“COVID-19”) AND (“agotamiento”))
	(“enfermera”) AND ((“COVID-19”) AND (“estrés”))

* Truncation operator that was used as a character truncation to right to find all forms of one word.

**Table 2 ijerph-19-04452-t002:** Summary of results.

Author, Year, Location	Design and Sample	Aim	Variables	Results	Quality, LE *, LR **
Nie et al., 2020 [41] China	Observational, descriptive, cross-sectional, descriptive study. 263 nurses.	To identify the prevalence and associated factors of psychological distress among frontline nurses during the COVID-19 outbreak.	Psychological distress, impact of the COVID-19 pandemic.	The risk factors that had the greatest impact on the nurses were: direct contact with patients with COVID-19, doubt about the efficacy of PPE, younger age, and stigmatization. Working overtime and changing work routine was not a risk factor for the nurses in this study. Loneliness, sadness, fear, and concern for their family members were triggered as responses. Faced with the stressors of the pandemic, nurses were more likely to develop positive strategies that were negatively related to psychological distress. Nurses who developed strategies such as overexertion in controlling coronavirus infection, avoidance, alcohol or drug use, and constructing false illusions were more likely to experience psychological distress.	STROBE 20/22 JBI *** 4b-C
Lyu et al., 2020 [40] China	Observational, descriptive, cross-sectional study. 216 nurses.	To explore how organizational identity and psychological resilience affect frontline nurses’ work engagement in coronavirus disease prevention and control 2019 (COVID-19) and to establish the relationship model based on these factors.	Level of resilience, organizational support, and work commitment.	The nurses in the study reported high levels of psychological resilience, which correlated positively with higher levels of perceived organizational support, leading to greater work engagement and quality of work. Resilience showed a mediating role between organizational support and nurses’ work engagement. Higher levels of resilience were also associated with having previously good emotional self-control and ability to adapt to challenges. The Chinese government issued a COVID-19 Disease Prevention and Control Plan, according to which Chinese hospitals were required to meet set standards and place special emphasis on developing organizational identity. Additional efforts were made to provide training and improve self-protection practices; protective measures were provided and evaluated, in turn improving nurses’ resilience and work engagement.	STROBE 19/22 JBI 4b-C
Cai et al., 2020 [29] China	Observational, descriptive, cross-sectional study. 1521 nurses.	To investigate psychological abnormality in healthcare workers struggling with the COVID-19 epidemic and to explore associations between social support, resilience, and mental health.	Influencing factors in psychological impact and the psychological responses they can trigger in nurses.	Risk factors influencing worse mental health outcomes were: younger age and lower family and social support, observing the increasing number of COVID-19 cases and deaths, and not having experience in public health emergencies. They tended to develop psychological abnormalities in interpersonal sensitivity, emotional distress, hostility, and obsessive–compulsive behaviors in response. The nurses in the sample presented moderate levels of resilience. Those who participated in previous epidemics had a significantly higher level of resilience compared to the others, which was associated with a better quality of the health interventions they provided. Perceived good social and organizational support, optimism, and tenacity were positively correlated with resilience and better mental health outcomes.	STROBE 20/22 JBI 4b-C
Cai et al., 2020 [28] China	Observational, descriptive, cross-sectional study. 534 health professionals: −248 nurses. −233 phisycians. −53 others.	To investigate the psychological impact and coping strategies of frontline medical personnel in Hunan province, adjacent to Hubei province, during the COVID-19 outbreak between January and March 2020.	Stressors and protective factors of healthcare professionals and psychological effects generated by the impact of the COVID-19 pandemic.	The factors associated with stress were: younger age, shortage of PPE, working in front of patients on the frontline, and seeing news about the evolution of COVID-19. Longer shifts and overtime were not a stressor. The predominant responses induced were: loss of control, feeling of vulnerability, nervousness, hostility, tension, concern for their safety and that of their families, and intention to leave their post. The availability of strict infection control guidelines, specialized equipment, recognition of their efforts by the hospital administration and the government, and the reduction in cases reported by COVID-19 were assumed to be protective factors. The most commonly used strategies to reduce stress were applying strict protective measures, seeking help from family and friends, having a positive attitude towards oneself and one’s work, and having a sense of humor. These strategies were mostly adopted by nurses compared to physicians and technicians. The least used strategy, in general, was to seek psychological support. Given the situation experienced by the nurses, in the survey, they expressed a number of interventions that could be made to promote their resilience and well-being: the availability of protective equipment, strict infection control guidelines, receiving specialized technical and emotional stress management training, and receiving support from hospital administration and government.	STROBE 19/22 JBI 4b-C
Leng et al., 2020 [36] China	Observational, descriptive, cross-sectional study. 108 nurses.	To quantify the severity of post-traumatic stress disorder (PTSD) symptoms and stress of nurses and to explore factors that influence their psychological health when caring for patients with COVID-19.	Regulatory factors of psychological impact on nurses and triggered responses. Degree of nurses’ resilience. Organizational interventions to foster resilience.	The main sources of stress included working in an isolated environment, shortage and prolonged use of PPE, intensity of the workload, contact with patients with COVID-19, lack of family support, and insufficient work experience. These factors generated responses in the nurses’ sleep disturbances, feeling of loneliness, guilt, and fear for their safety. The nurses in the study had moderate resilience scores. Even the resilient nurses experienced some degree of mental distress, although significantly less compared to the others. Perceiving greater organizational support through leadership rounds; providing breaks, psychological support, training, and coaching; and a reduced workday were associated with better resilience scores.	STROBE 20/22 JBI 4b-C
Luo et al., 2020 [39] China	Systematic review and meta-analysis. 62 studies.	To assess the updated psychological and mental impact of the COVID-19 pandemic among health care workers.	Risk and protective factors related to psychological impact.	Among health professionals, factors such as being a woman, a nurse, working on the frontline in direct contact with COVID-19 patients, working in the most affected area, not having adequate protective measures, being aware of news about the evolution of COVID-19, and having a lack of training and family support were additionally associated with greater psychological distress.	AMSTAR-2 RATING High JBI 3a-A
Cunill et al., 2020 [45] Spain	Observational, descriptive, cross-sectional study. 1452 nurses.	To describe the physical and psychological symptoms in health care workers caring for patients with COVID-19.	Stressors and psychological and physical responses triggered by psychological impact on nurses.	Shortage of protective material, work overload, working in isolated environments, prolonged use of PPE, being a woman, being a nurse, and having children were detected as stressors. They triggered responses such as uncertainty due to not knowing if they have the disease, helplessness, discomfort, perception of not being able to perform their professional duties effectively, all of which give rise to physical symptoms, such as headaches, arms, legs, back, and precordial pain, fatigue and insomnia, gastrointestinal problems, decreased appetite, dyspnea, dizziness, and/or problems in sexual intercourse.	STROBE 21/22 JBI 4b-C
Luceño-Moreno et al., 2020 [38] Spain	Observational, descriptive, cross-sectional study. 1422 health professionals: −826 nurses. −428 physicians. −168 nursing assistants.	Analyzing post-traumatic stress, anxiety, and depression during the COVID-19 pandemic.	Influencing factors in the psychological impact of the COVID-19 pandemic on nurses and induced responses.	The risk and stress factors were: being a woman, being younger, having less work experience, working in a hospital, being a nurse, having a 12- or 24-hour on-call shift, and living with people at risk. The main responses were: uncertainty and fear of the possibility of being infected and being able to pass it on to family members. The levels of resilience of the health care workers evaluated were moderate. Resilience was negatively correlated with emotional exhaustion and depersonalization and positively influenced the mental health of healthcare workers. Having a graduate or doctoral degree was associated with higher levels of resilience.	STROBE 20/22 JBI 4b-C
Giusti et al., 2020 [32] Italy	Observational, descriptive, cross-sectional study. 330 health professionals: −140 physicians. −86 nurses. −38 nursing assistants. −67 others.	To assess the prevalence of burnout and psychopathological conditions in health professionals working in a healthcare institution in northern Italy and to identify sociodemographic, occupational, and psychological predictors of burnout.	Risk factors and their relationship with psychological symptoms.	Predictors of greater psychological impact were: longer working hours, previous psychological comorbidities, fear of infection, feelings of isolation, less perceived support from friends, female gender, being a nurse, age, working in the hospital, and being in contact with patients with COVID-19.	STROBE 20/22 JBI 4b-C
Labrague y de los Santos 2020 [35] Filipinas	Observational, descriptive, cross-sectional study. 325 nurses.	To examine the relative influence of personal resilience, social support, and organizational support on the reduction in COVID-19 anxiety in frontline nurses.	Regulatory factors of psychological impact in frontline nurses and triggered responses. Nurses’ level of resilience, organizational support, and social support.	The most significant risk factor for the nurses’ discomfort due to the impact of the pandemic was not feeling prepared for the management of patients with COVID-19 and perceiving little organizational and social support. The most frequent psychological and physical behavioral responses were: tonic immobility, insomnia, dizziness, loss of appetite, and abdominal discomfort. Nurses presented moderate levels of resilience. Better resilience scores were associated with greater perceptions of social and organizational support, and in turn, with reduced anxiety in nurses.	STROBE 20/22 JBI 4b-C
Kilinç y Çelik. 2020 [34] Turkey	Observational, descriptive, cross-sectional study. 370 nurses.	To determine the relationship between social support and levels of psychological resilience perceived by nurses in Turkey during the coronavirus disease pandemic-2019 (COVID-19).	Sociodemographic variables. Level of social support. Level of resilience.	The nurses’ levels of resilience were moderate. A significant positive directional relationship was observed between economic status, age, job consolidation, working conditions improved by the hospital, social and family support perceived by the nurses, and their level of psychological resilience. Being in contact with COVID-19 patients had a negative influence on the nurses’ levels of resilience.	STROBE 18/22 JBI 4b-C
Zhang et al., 2020 [43] China	Observational, descriptive, cross-sectional study. 110 nurses.	To identify stressors and burnout among frontline nurses caring for COVID-19 patients in Wuhan and Shanghai and explore perceived effective moral support strategies.	Stressors and responses to them in nurses who cared for patients with COVID-19.	The most frequent stressors among the nurses were: lack of family support, work experience, time spent working in isolation rooms, prolonged use of PPE, and younger age. Among the responses triggered by this were: loneliness, guilt, fear of separation from their families, uncertainty, fear of infection, discomfort due to prolonged use of PPE, and concern about providing poor nursing care. The strategies adopted by the nurses to cope with stress were: acquiring training, seeking information on mental health, adopting a positive attitude, participating in health-promoting activities, and practicing relaxation techniques. The least used were those related to seeking professional psychological support. According to the nurses, the most effective main support interventions that could be adopted by the hospital to contribute to reducing stress and improving their resilience and well-being were: support from supervisors, provision of sufficient material, clear instruction on treatment procedures and on COVID-19, offering sufficient time off, professional promotion, and offering psychological services.	STROBE 18/22 JBI 4b-C
Kim et al., 2021 [33] USA	Observational, descriptive, cross-sectional study. 320 enfermeras.	To examine the impact of various factors affecting nurses’ mental health during the COVID-19 pandemic.	Influencing factors in nurses in the face of the COVID-19 pandemic in relation to their stress level.	The level of stress, anxiety, and distress perceived by the nurses was moderate–severe, higher than that estimated before the pandemic. Predictors of stressors were: patient care with COVID-19, isolation, younger age, fewprofessional experience, and poor family functioning. Nurses reported moderate levels of resilience. Those with high levels were two to six times less likely to have poor mental health. Organizational support, good family functioning, and spirituality were factors that positively influenced the level of resilience, making nurses less likely to have poor mental health.	STROBE 18/22 JBI 4b-C
Rodríguez-Vega et al., 2020 [47] Spain	Qualitative study: exploratory research with a post-intervention evaluation. 150 health professionals.	Implementing a mindfulness-based intervention for frontline healthcare workers during the COVID-19 outbreak in a general public hospital in Madrid.	Attendance at the session. Job position. Perceived usefulness.	More than 3000 sessions were carried out by professionals of Intensive Care Units, COVID-19 Medical Units, and Emergency Services. It was well accepted specifically by nurses and nursing assistants; physicians presented more rejection and were the professionals who attended the least. The intervention was evaluated in the short term and qualified as very useful for reducing stress in frontline health workers, favoring their resilience and reflecting data on feasibility, usefulness, and safety.	ICROMS 17 JBI 3-C
Pollok et al., 2020 [44] UK	Systematic review. 16 studies applied to health professionals working on the frontline during epidemics or disease outbreaks.	To assess the effects of interventions aimed at supporting the resilience and mental health of frontline health and social care professionals during and after a disease outbreak, epidemic, or pandemic, as well as the barriers and facilitators to implementing them.	Interventions to promote resilience: related to working conditions; to support basic daily needs; psychological support. Barriers and guidelines for implementation.	Among the most prominent interventions to foster resilience of health professionals in other outbreaks were: those related to working conditions (regular breaks, shorter working hours, team meetings, relaxation/recreation areas in workplaces, provision of epidemic training for professionals, training of professionals in helping patients and self); to support basic daily needs (food, rest); and those of psychological support (online, group therapies, etc.). A number of barriers were identified, such as lack of awareness of the needs of frontline workers by organizations and limited resources and guidelines, for the successful implementation of resilience-related interventions.	AMSTAR-2 RATING High JBI 3b-B
Afshari et al., 2021 [26] Irán	Observational, descriptive, cross-sectional study. 387 nurses.	To determine the resilience score and demographic predictors among nurses working in hospitals involved with COVID-19.	Level of resilience. Demographic factors: age, gender, work experience, education level, marital status, and offspring. Sports activity, hospital classification, and degree of labor consolidation.	Nurses’ levels of resilience were low–moderate. Older age, educational level, experience, and a consolidated job position were positively correlated with resilience. Having children, being female, and not exercising were associated with lower resilience scores.	STROBE 18/22 JBI 4b-C
Ou et al., 2021 [42] China	Observational, descriptive, cross-sectional study. 92 nurses.	Evaluated the impact of supportive interventions on resilience and self-rated psychopathological symptoms of 92 nurses working in the COVID-19 isolation ward.	Psychopathological symptoms: somatization, obsessive–compulsive behaviors, problems in developing interpersonal sensitivity, depression, anxiety, hostility, phobia, paranoid ideation, and psychoticism. Level of resilience.	The nurses presented high levels of resilience, which were markedly higher than those obtained in other studies. Past experiences in public health emergencies and interventions developed by the hospital to foster organizational and social support, such as training of nurses to better manage the psychological problems of their patients and training on diagnostic guidelines and treatment of COVID-19 before entering the isolation ward. They were given priority access to PPE, a good working environment was fostered, flexible shifts were implemented according to work intensity, and a support team was established to protect the ward workforce. This notably influenced higher resilience scores in the nurses, considerably decreasing psychopathological symptoms, improving their sleep and eating quality, and leading to better mental health in nurses and quality results in their work.	STROBE 20/22 JBI 4b-C
Lorente et al., 2021 [37] Spain	Observational, descriptive, cross-sectional study. 421 nurses.	To analyze the cross-sectional effect of sources of stress during the peak of the COVID-19 pandemic on nurses’ psychological distress, focusing on the mediating role of coping strategies, including problem-focused, emotion-focused, and resilience.	Stressors of nurses on the frontline of care for patients with COVID-19 and their response to those stressors.	Identified stressors: work overload, insufficient preparation to cope with work demands, and lack of support provoked fear of infection and death in nurses. Resilience is negatively and significantly related to psychological distress. Resilience can play an important role in improving mental health, but it will only reach higher levels and be a relevant mediator in the stressor–psychological distress relationship when the stressors have induced emotion- and problem-focused strategies. Individually, emotion-focused strategies combat psychological distress and are related to the development of resilience. Problem-focused strategies alone are related to higher levels of psychological stress and distress in nurses. Nurses who experience stress due to insufficient preparation and fear of contagion do not implement adequate coping strategies and thus will not be able to acquire resilience.	STROBE 20/22 JBI 4b-C
Rieckert A et al., 2021 [46] Nether- lands	Scoping review. 73 articles.	To explore how to develop and maintain the resilience of frontline healthcare professionals exposed to the working conditions of the COVID-19 outbreak.	Interventions before or during the outbreak to build nurse resilience.	Recommendations prior to the outbreak to promote resilience included: optimal provision of education, resilience training, and interventions to create a sense of preparedness for clinical practice. Recommendations during the outbreak consisted of: enhancing resilience through adequate provision of information, psychosocial support, and treatment; monitoring the health status of practitioners; and utilizing various forms and contents of psychosocial support.	AMSTAR-2 RATING Moderate JBI 4a-C
Balay-Odao et al., 2021 [27] Saudi Arabia	Observational, descriptive, cross-sectional study. 281 nurses.	To determine predictors of hospital preparedness in the management of patients with COVID-19 and psychological burden and resilience among clinical nurses in addressing the COVID-19 crisis in Saudi Arabia.	Influencing factors in the psychological burden of nurses.	The relevant factors that were revealed as predictors of psychological burden in nurses were: being female, living with family members, being married, working in the emergency area or isolation room, not being trained for the integral management of the patient with COVID-19, and having a low economic level. The nurses perceived high hospital preparedness to correlate with moderate–high levels of resilience, decreasing the mean score of anxiety, stress, and depression. Factors such as age, experience, educational level, degree of trust in authorities, and perceived social and family support fostered resilience and reduced nurses’ psychological burden. Hospital preparation, prevention, control, management, and containment of COVID-19, and government, social, and family support for nurses were relevant aspects that favored resilience. Having a learning attitude, positive thinking, and acquiring updated training were resilience strategies for nurses to adapt psychologically in a more optimal way.	STROBE 18/22 JBI 4b-C
Del Pozo-Herce et al., 2021 [31] Spain	Observational, descriptive, cross-sectional study. 605 health professionals: −63.14% nurses. −36.28% nursing assistants.	To determine the psychological impact of the SARS-CoV-2 virus on nursing professionals working in the Rioja Health Service (Spain).	Stressors and protective factors. Perceived emotions.	Influencing factors such as being a woman, being younger and less experienced, not having a consolidated contract, having dependent family members, and being in contact with COVID-19 patients gave rise to concerns about fear of becoming infected or infecting their loved ones, making mistakes, as well as negativism, emotional destabilization, and sadness for not providing adequate physical and/or psychoemotional care to the patient’s needs. The strategies developed by the health professionals were: seeking help from prepared materials, bibliographies, and psychological resources available online. A very small percentage sought specialized external help. Women used fewer coping strategies.	STROBE 20/22 JBI 4b-C
De Pinho et al., 2021 [30] Portugal	Observational, descriptive, cross-sectional study. 820 nurses.	To explore the association between mental-health-promotion strategies used by nurses during the COVID-19 outbreak and their symptoms of depression, anxiety, and stress.	Strategies to promote mental health.	Healthy eating, physical activity, resting between shifts, maintaining social contacts, verbalizing feelings/emotions, and spending less time seeking information on COVID-19 were strategies developed by nurses that were associated with better mental health.	STROBE 19/22 JBI 4b-C

* LE: level of evidence. ** LR: level of recommendation; *** JBI: Joanna Bridge Institute.

## Data Availability

Not applicable.

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
