# Peer review of "Modulating Elements of Nurse Resilience in Population Care during the COVID-19 Pandemic"

_ijerph, 2022, doi:10.3390/ijerph19084452_

Round 1

Reviewer 1 Report

Dear Authors! The choice of topic is topical, relevant, and interesting. The methodology is good, but the precise definition of the exclusion criteria is missing, and the corresponding table is not sufficiently informative (no precise definition of inclusion and exclusion criteria). The conclusion is too general, and the opponent lacks concrete proposals and solutions that are forward-looking in the light of the results.

Author Response

Thank you very much for taking the time to indicate remarks, recommendations and suggestions to improve the article. They were really helpful. We have included them in the manuscript. We believe that the quality of the article has improved greatly thanks to your suggestions.

  1. The methodology is good, but the precise definition of the exclusion criteria is missing, and the corresponding table is not sufficiently informative (no precise definition of inclusion and exclusion criteria).

Thank you very much for your suggestion, the inclusion and exclusion criteria have been clarified following your recommendation. (Page 3, lines 110-120)

  1. The conclusion is too general, and the opponent lacks concrete proposals and solutions that are forward-looking in the light of the results.

Thank you very much, the findings of the study including forward-looking recommendations derived from the results of this review have been finalised. (Page 23, lines 388- 410)

Reviewer 2 Report

Thank you for submitting your manuscript, I will now make some recommendations for improvement it:

  • It has been quite some time since May 2021 and I think perhaps the search should have been done again to check if there are new manuscripts to be included.
  •  I think there is a lack of review in databases such as Scopus and PsycINFO.
  • Figure 1 has the word “estudios” in Spanish
  • STROBE guidelines are paper reporting guidelines and are not to be used as study quality/bias assessment tools.
  • The author contribution needs to be filled in. 

Author Response

Thank you very much for taking the time to indicate remarks, recommendations and suggestions to improve the article. They were really helpful. We have included them in the manuscript. We believe that the quality of the article has improved greatly thanks to your suggestions.

  1. It has been quite some time since May 2021 and I think perhaps the search should have been done again to check if there are new manuscripts to be included.

Thank you very much for your indication, indeed a considerable period of time has passed and it would be interesting to follow the recommendation, but we consider that the results of the study have been collected in the most critical period of time of the pandemic and when more publications of interest emerged on the subject. After carrying out a small search we have seen that the current publications focus on the post-pandemic period, and not on the critical period which is the one we are interested in, so we have taken the decision to maintain the search period, assuming that as you say, some studies may have been published later.

  1. I think there is a lack of review in databases such as Scopus and PsycINFO.

We are very grateful for your appreciation, indeed, some psychosocial articles may have been missed, but we were looking for a clinical and nursing approach, so the search was focused on the health science databases that were considered most relevant.

We add these two suggestions in the limitations of the review (page 23, lines 363-367).

  1. Figure 1 has the word “estudios” in Spanish

Thank you very much for your feedback, we have corrected the error and updated the figure to the latest format (page 6).

  1. STROBE guidelines are paper reporting guidelines and are not to be used as study quality/bias assessment tools.

Indeed, the STROBE guidelines should not be used as tools for quality/bias assessment. The STROBE, CASPE and PRISMA tools have been replaced by the ICROMS tool that assesses quality and risk of bias (page 4, lines 147-153 and table 2, page 8).

Reviewer 3 Report

IJERPH

 March 13th, 2022

Manuscript ID: ijerph-1614462

Title: "Modulating elements of nurse resilience in population care during the COVID-19 pandemic".

General comments:

Thank you for the opportunity to review this timely article on an important and topic on modulating elements of nurse resilience in population care during the COVID-19 based on Dorothy Johnson's Behavioral Model.

The authors carried out a systematic review aiming to analyze the resilience of care nurses to the psychological impact of 22 the COVID-19 pandemic.

Although the study is well-written as well as is quite relevant, I have many concerns about the methodology od this manuscript that is fragile.

Please find below some comments, suggestions in order to strengthen the potential contribution of this topic in any revision the author(s) might undertake. 

METHODS

This is the most complicated section of the article as it has many limitations and needs further work.

First, this review would be more suitable for a scoping review than a systematic review guided by the PRISMA guidelines (because many of the PRISMA checklist items were not met).

According to the Cochrane Collaboration, no time period or language should be limited to SYSTEMATIC REVIEWS (due to publication bias); However, this is permitted in an integrative review or a scoping review in accordance with the JBI.

I would like to know if this systematic review was registered with PROSPERO or OSF? Please report this at the beginning of the method and justify if it has not been recorded (this is one of the items in the PRISMA Checklist Statement 2020)

Which acronym was used to prepare the guiding question: PICO, PECO, PCC, SPIDER? Inform this in the method as well as make it explicit in the table format addressing the inclusion and exclusion criteria that must be aligned with the acronym. Once again, for your type of question, the most suitable acronym is PCC (Population, concept, context), that is, further ratifying that this should be a scope review and not a systematic review.

According to the Cochrane Handbook for intervention studies, the 3 essential bases should be used minimally, ie: MEDLINE/PubMed; Embase and Cochrane Libary. Ideally, at least one specific database is recommended (and in the case of this study I would recommend CINAHL - which is the international nursing database as well as PsycINFO) in addition to at least one complementary database (Web of Science, SCOPUS, Science Direct, etc). The results (total articles retrieved) were certainly influenced by the restricted number of bases, mainly, by the delimitation of time period and language. As this is a systematic review it is important to cover the literature as much as possible in order to avoid publication bias. Furthermore, gray literature was not searched. Why did the authors choose not to include gray literature? This needs to be clear in the method and even go to the limitations section of the study. In addition, both Cochrane Handbook 2021 and JBI Manual 2020 recommend searching the gray literature.

The Search Strategy is not sensitive enough to capture eligible articles. By the way, was any database specialist or librarian consulted to carry out this strategy?

Well, if you are using LILACS, it is important to use the controlled descriptors (DeCS terms), with the combination of synonyms  in the four languages that this database indexes the DECs: Portuguese, English, Spanish and French.

I strongly recommend expanding the search on Embase, CINAHL and PsycINFO.

Critical appraisal:

The authors claim to have performed the risk of bias assessment, but this was not done. It is wrong to use PRISMA recommendations or any other Equator checklist (STROBE or CASPe or COREC) to assess risk of bias. It is important to differentiate, evaluation of the study report of risk of bias assessment, which are completely different things.

For risk of bias assessment there are specific tools by design as recommended by the Cochrane Collaboration Handbook or by the JBI.

If you are going to keep the systematic review design, you will have to change the tools from each design, as the Equator checklist does not assess the risk of bias in a systematic review, but the study report.

For example, for observational studies, the options for specific tools for assessing methodological quality are: Appraisal tool for Cross-Sectional Studies (AXIS tool) OR

The Agency for Healthcare Research and Quality (AHRQ) OR Joanna Briggs Institute (JBI) JBI Critical Appraisal Checklist for Studies Reporting Prevalence Data.

References

Downes MJ, Brennan ML, Williams HC, Dean RS. Development of a critical appraisal tool to assess the quality of cross-sectional studies (AXIS). BMJ Open. 2016;6(12):e011458.

Agency for Healthcare Research and Quality. Methods guide for effectiveness and comparative effec- tiveness reviews; AHRQ Publication 10(14)-EHC063-EF. Rockville, MD: Agency for Healthcare Research and Quality;2014.

Critical Appraisal Tools - JBI. 2017. Disponível em http://joannabriggs.org/research/ critical-appraisal-tools.html 

For systematic review studies, the methodological tool should be AMSTAR-2:

Shea BJ, Grimshaw JM, Wells GA, Boers M, Andersson N, Hamel C, et al. Development of AMSTAR: a measurement tool to assess the methodological quality of systematic reviews. BMC Med Res Methodol. 2007;7:10.

Or you can also evaluate using JBI's specific tools for each design, if your systematic review is guided by the JBI Manual 2020

RESULTS

Figure 1. The PRISMA FLOWCHART, needs to be replaced by the current one published this year in The BMJ - DOI: 10.1136/bmj.n71  (PRISMA statement 2020) –

http://prisma-statement.org/prismastatement/flowdiagram.aspx

Item 16a of the PRISMA checklist 2020 is precisely the study selection and is the first part of the results and not methods section. Figure 1 can't be in the method section (needs to be consistent with the PRISMA guideline you're following, right?)

 DISCUSSION

I missed some studies in the area that were not mentioned

Limitations of the study

The authors must recognize the several limitations of this review.

I emphasize that the article is interesting, however, it needs to be better worked mainly the methods section.

Dr. Luís Carlos Lopes-Júnior - Ad-hoc Consultant and Review Studies Specialist

Author Response

Thank you very much for taking the time to indicate remarks, recommendations and suggestions to improve the article. They were really helpful. We have included them in the manuscript. We believe that the quality of the article has improved greatly thanks to your suggestions.

  1. According to the Cochrane Collaboration, no time period or language should be limited to SYSTEMATIC REVIEWS (due to publication bias); However, this is permitted in an integrative review or a scoping review in accordance with the JBI.

Thank you very much, indeed the guidelines of the Cochrane Collaboration are not scrupulously followed, we include the publication bias in the limitations of the study (page 23, lines 363-367).

  1. I would like to know if this systematic review was registered with PROSPERO or OSF? Please report this at the beginning of the method and justify if it has not been recorded (this is one of the items in the PRISMA Checklist Statement 2020)

Interesting observation. No, this review was not recorded due to the rush to collect data in the midst of the pandemic and the desire to share our findings quickly. We have included this at the beginning of the method (page 3, line 106).

  1. Which acronym was used to prepare the guiding question: PICO, PECO, PCC, SPIDER? Inform this in the method as well as make it explicit in the table format addressing the inclusion and exclusion criteria that must be aligned with the acronym. Once again, for your type of question, the most suitable acronym is PCC (Population, concept, context), that is, further ratifying that this should be a scope review and not a systematic review.

Thank you very much, throughout the study we tried to follow the PRISMA guidelines for a systematic review. The guiding question was prepared following the acronym PICO, a section on methods is included with all the considerations (page 3, lines 98-105).  

  1. According to the Cochrane Handbook for intervention studies, the 3 essential bases should be used minimally, ie: MEDLINE/PubMed; Embase and Cochrane Libary. Ideally, at least one specific database is recommended (and in the case of this study I would recommend CINAHL - which is the international nursing database as well as PsycINFO) in addition to at least one complementary database (Web of Science, SCOPUS, Science Direct, etc). The results (total articles retrieved) were certainly influenced by the restricted number of bases, mainly, by the delimitation of time period and language. As this is a systematic review it is important to cover the literature as much as possible in order to avoid publication bias. Furthermore, gray literature was not searched. Why did the authors choose not to include gray literature? This needs to be clear in the method and even go to the limitations section of the study. In addition, both Cochrane Handbook 2021 and JBI Manual 2020 recommend searching the gray literature. I strongly recommend expanding the search on Embase, CINAHL and PsycINFO.

We are very grateful for your appreciation, indeed, some psychosocial articles may have been missed, but we were looking for a more clinical and nursing approach and included the health science databases that were considered most relevant. We include this suggestion in the limitations of the study (page 23, lines 363-367).

  1. The Search Strategy is not sensitive enough to capture eligible articles. By the way, was any database specialist or librarian consulted to carry out this strategy?

Thank you very much for your appreciation. No, we did not have access to a librarian or documentalist for consultation. However, the search and selection process involved researchers with extensive experience and numerous published systematic reviews, including psychometric reviews, so we are confident that the main results are included in our study.

  1. Well, if you are using LILACS, it is important to use the controlled descriptors (DeCS terms), with the combination of synonyms in the four languages that this database indexes the DECs: Portuguese, English, Spanish and French.

Thank you very much. The guidelines you recommend for searching the LILACS database were indeed followed.

  1. The authors claim to have performed the risk of bias assessment, but this was not done. It is wrong to use PRISMA recommendations or any other Equator checklist (STROBE or CASPe or COREC) to assess risk of bias. It is important to differentiate, evaluation of the study report of risk of bias assessment, which are completely different things.

Indeed STROBE guidelines should not be used as tools for quality/bias assessment. The STROBE, CASPE and PRISMA tools have been replaced by the ICROMS tool that assesses quality and risk of bias (page 4, lines 147-153 and table 2, page 8).

  1. Figure 1. The PRISMA FLOWCHART, needs to be replaced by the current one published this year in The BMJ - DOI: 10.1136/bmj.n71 (PRISMA statement 2020) –http://prisma-statement.org/prismastatement/flowdiagram.aspx

Thank you very much, the flow chart has been replaced by the model published in the PRISMA 2020 statement (page 6).

  1. Item 16a of the PRISMA checklist 2020 is precisely the study selection and is the first part of the results and not methods section. Figure 1 can't be in the method section (needs to be consistent with the PRISMA guideline you're following, right?)

Indeed, thanks for the comment, figure 1 has been moved to the results section (page 6).

  1. The authors must recognize the several limitations of this review.

Thank you for your appreciation, following your recommendations, the limitations

Round 2

Reviewer 3 Report

IJERPH

March 28th, 2022

Manuscript ID: ijerph-1614462-R1

Title: "Modulating elements of nurse resilience in population care during the COVID-19 pandemic".

I thank the authors for the answers to the questions raised, which were appropriate. And now the paper is better presented.

However, I still suggest few adjustments to the manuscript:

Summary: Fix PubMed to (Medline/Pubmed) item in Table 1.

Page 3. Explain in the same way as in the reply letter why the protocol was not registered. “The protocol for this review was not registered due........”

Table 1. This table should contain only the information of the search key used. (Columns 1 and 2). It is not necessary to include in the method both the obtained and selected articles (the selection process is results according to PRISMA and does not fit here). I recommend removing column 3 and 4.

Page 4.

Briefly describe the levels of evidence of effectiveness according to the Joana Briggs Institute.

Briefly describe the ICROMS tool and AMSTAR used to assess the risk of bias, as well as the interpretation of these tools.

Member of Reviewer Board of the IJERPH – Dr. Luís Carlos Lopes-Junior

Author Response

We greatly appreciate the recommendations, which we believe significantly improve the manuscript.

  1. Summary: Fix PubMed to (Medline/Pubmed) item in Table 1.

Thank a lot, We change it.

  1. Page 3. Explain in the same way as in the reply letter why the protocol was not registered. “The protocol for this review was not registered due........”

Thanks, We have added the following in the text:

The protocol for this review was not registered, due to the need to collect data in the midst of the pandemic and the desire to share our findings quickly.

  1. Table 1. This table should contain only the information of the search key used. (Columns 1 and 2). It is not necessary to include in the method both the obtained and selected articles (the selection process is results according to PRISMA and does not fit here). I recommend removing column 3 and 4.

Thanks, We have removed the columns you suggest.

  1. Page 4. Briefly describe the levels of evidence of effectiveness according to the Joana Briggs Institute.

OK. Modified

  1. Briefly describe the ICROMS tool and AMSTAR used to assess the risk of bias, as well as the interpretation of these tools.

OK. Modified.

Additionally:

After much deliberation, We have finally decided to maintain the evaluation of cross-sectional articles with the STROBE guide, since we have found that it is commonly found in many systematic reviews published in IJERPH. Therefore, We do not accept the suggestion of reviewer 3 in the previous review; We believe that the strobe evaluation is methodologically more suitable than the one performed with icroms. We assume that strobe does not perform bias assessment and have noted this in the study limitations.

We add in limitations section:

On the other hand, given that during the worst moments of the COVID-19 pandemic, scientific production was necessarily rapid and agile, it is logical that many cross-sectional observational articles were published, which we have decided to include in the study despite the low evidence they provide. To evaluate them we chose the STROBE tool, which, although it does not evaluate bias, is one of the tools used in Cochrane and is commonly used in published reviews, also of a systematic type.
